# The First Fossil Coelacanth from Thailand

Lionel Cavin [1,*], Haiyan Tong [2], Eric Buffetaut [2,3], Kamonlak Wongko [4], Varavudh Suteethorn [5] and Uthumporn Deesri [6]

1 Department of Geology and Palaeontology, Natural History Museum of Geneva, 1208 Geneva, Switzerland
2 Palaeontological Research and Education Centre, Mahasarakham University, Khamrieng, Kantharawichai District, Maha Sarakham 44150, Thailand
3 CNRS (UMR 8538), Laboratoire de Géologie de l'Ecole Normale Supérieure, 24 rue Lhomond, 75231 Paris Cedex 05, France
4 Mineral Resources Office Region 1, Department of Mineral Resources, 414 moo 3, Sala, Ko Kha, Lampang 52130, Thailand
5 Dinosaur Research Unit, Mahasarakham University, Khamrieng, Kantharawichai District, Maha Sarakham 44150, Thailand
6 Department of Biology, Faculty of Science, Mahasarakham University, Khamrieng, Kantharawichai District, Maha Sarakham 44150, Thailand
* Correspondence: lionel.cavin@ville-ge.ch

**Abstract:** Mawsoniidae is a family of coelacanths restricted to the Mesozoic. During the Cretaceous, mawsoniids were mainly represented by the *Mawsonia/Axelrodichthy* complex, long known to be from western Gondwana only (South America and Africa). This apparent biogeographical distribution then faded following the discovery of representatives in the Late Cretaceous of Laurasia (Europe and North America). We report here the presence, in the Lower Cretaceous site of Kham Phok, NE Thailand, of an angular bone referred to the *Mawsonia/Axelrodichthys* complex. A comparison with angulars referring to both genera found in various regions of the world between the Late Jurassic and the Late Cretaceous indicated that the distinctions between these genera, and even more so between their constituent species, are unclear. This discovery is further confirmation of the very slow morphological evolution within this lineage, which may explain why their evolutionary history appears to be disconnected, at least in part, from their geographical distribution over time.

**Keywords:** Actinistia; Mawsoniidae; paleobiogeography; angular; Khorat Plateau; Early Cretaceous; *Axelrodichthys*; *Mawsonia*





## 1. Introduction

Mawsoniidae is a Mesozoic family of coelacanths, sister to the extant Latimeriidae. They occupied a wide range of aquatic environments, from fresh to marine waters, but with a preference for continental environments, especially during the Cretaceous. First discovered at the turn of the 20th century in the Early Cretaceous of Brazil [1], mawsoniid fossils were later found in various freshwater and euryhaline deposits in that country [2–5] and Uruguay [6,7]. Meanwhile, mawsoniid remains have been described from several Early to 'mid' Cretaceous localities in North Africa [8–13] and then in Central Africa [14,15].

When restricted to the Cretaceous, the palaeogeographical pattern reconstructed from these early discoveries was simple, namely one or more vicariance events on western Gondwana associated with the opening of the South Atlantic Ocean. All Cretaceous occurrences have been referred to a pair of sister genera, *Mawsonia/Axelrodichthys*, dubbed the *Mawsonia/Axelrodichthys* complex by Forey [16], each containing several species. Due to the often fragmentary nature of the discoveries, the delimitations of species and genera are still imprecise, preventing the construction of a well-supported phylogeny within the complex.

Over the past two decades, however, the western Gondwana model has been challenged with the discovery of mawsoniid remains in the Upper Cretaceous of Madagascar [17] and southern France [18–20], leading to a hypothesis of dispersal events from Gondwana to these peripheral landmasses [21]. Secondly, very recent discoveries in the 'mid' Cretaceous of North America [22] and Europe [23] challenged the dispersal hypothesis, suggesting that the origin of the *Mawsonia/Axelrodichthys* complex may be older and that its biogeographical distribution could be associated with the break-up of Pangaea.

Here, we record the presence of an indeterminate mawsoniid found in the basal Cretaceous of northeastern Thailand on the basis of an isolated angular bone discovered in the Kham Phok site, Mukdahan province. This unexpected occurrence sheds new light on the evolutionary history of the *Mawsonia/Axelrodichthys* lineage, confirming its ancient origin and wide palaeogeographical distribution associated with the breakup of Pangaea. Incidentally, this discovery is a new example of the very conservative morphology of the species constituting the *Mawsonia/Axelrodichthys* complex, confirming the 'living fossil' status of this lineage of coelacanths.

## 2. Geological and Paleoenvironmental Settings

The single specimen, comprising an angular associated with a skull roof bone referable to an undetermined mawsoniid, was found in the Kham Phok locality, Khamcha-i district, Mukdahan province, located in the Phu Kradung Formation (Figure 1). The Phu Kradung Formation, a series of sandstone bars alternating with silty pedogenetic horizons corresponding to fluvial deposits, outcrops along the Phu Phan mountain range, which crosses the Khorat plateau, and along the western edge of this plateau. The Phu Kradung Fm. is the lowest formation of the Khorat Group, which is a succession of five continental formations ranging from the Upper Jurassic, i.e., the lower part of the Phu Kradung, to the Aptian with the Khok Kruat Formation. Stratigraphically, the Kham Phok site is located in the upper part of the Phu Kradung Formation. Based on the assemblages of hybodont shark teeth, of crocodiles and of turtles, a basal Cretaceous age was suggested [24,25].

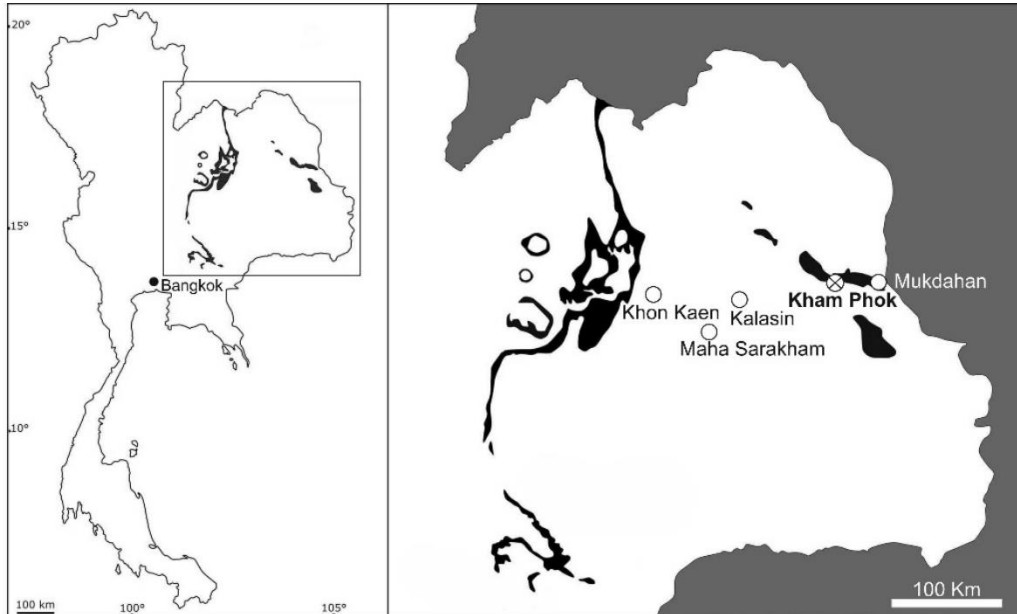

**Figure 1.** Location of the Kham Phok site; black color represents the distribution of the Phu Kradung Formation on the Khorat Plateau.

The vertebrate assemblage from Kham Phok comprises at least four hybodont sharks [24], a single juvenile specimen referred to the ginglymodian ray-finned fish *Thaiichthys buddhabutrensis* [26], the giant cryptodiran turtle *Basilochelys macrobios* [27], the pholidosaurid crocodile *Chalawan thailandicus* [28] and a sinraptorid [29].

## 3. Materials and Methods

The specimen (PRC 160) was discovered in 2005 at Kham Phok during a field trip conducted by the Thai–French team. It consists of two ossifications strongly attached to each other by the matrix, but not in an anatomical association, which probably belong to the same individual. The specimen was mechanically prepared by one of us (HT) using mounted needles. The bones were not separated from each other because of the strong attachment by the matrix and because of the fragility of the specimen.

The anatomical nomenclature of the angular follows Cavin et al. [22], except that we consider the attachment surface with the surrounding bones as "sutural" surfaces rather than "articular" surfaces because no movement was possible between the bones constituting the lower jaw.

## 4. Results

### 4.1. Systematic Paleontology

Actinistia Cope, 1871 [30]
Latimerioidei Schultze, 1993 [31]
Mawsoniidae Schultze, 1993 [31]
Mawsoniidae indeterminate
Referred material: A left angular associated with a dermal skull bone (supraorbital?) (PRC 160) from the Kham Phok fossil site.
Locality and horizon: Kham Phok, Khamcha-i district, Mukdahan Province, upper part of the Phu Kradung Formation, basal Cretaceous.

### 4.1.1. Description

- The state of preservation of the left angular PRC 160 (Figure 2) is fairly good, except for some breaks in its mid-length and along its ventral side. Moreover, matrix that cannot be removed because of fragility of the bone obscures some details of its medial (lingual) face. A dermal skull bone, likely belonging to the same individual, is still attached with matrix to the posterior part of the medial side. Although incompletely preserved, the general outline of the angular corresponds to a rather shallow bone, with its length a little more than four times its depth. The posterodorsal margin of the bone is poorly preserved, and its anterodorsal margin is almost straight. The coronoid eminence, located at the anterior third of the bone, is very slightly inclined forward. The sutural surface with the principal coronoid is small, but well defined. The lateral (labial) side is ornamented with a dense pattern of reticulated ridges oriented along the anteroposterior axis of the bone in its mid-depth and oriented toward the coronoid eminence in the anterodorsal region of the bone. The overlap surface for the dentary is visible as a slight concavity dug in the labial side along the anterodorsal margin, extending almost to the coronoid eminence. The ornamentation on this sutural surface, composed of anastomosed ridges smaller than those on the rest of the bone, is more noticeable than that in other mawsoniids. The ventral margin of the bone is inwardly curved, and this region is almost devoid of ornamentation. The ventral side of the angular is poorly preserved, obscuring the pattern of the mandibular sensory canal. A large posterior opening corresponding to the entrance of the mandibular sensory canal is present, visible in lateral and ventral views (Figure 2). In the ventral view, several pores are visible in the posterior third of the bone, but it is difficult to determine their number and shape. More anteriorly, the pores are likely present but not visible because of preservation. On the lingual face, the adductor fossa is well developed and marked by a pronounced ridge along its ventral margin. The longitudinal fossa, located on the posterior part of the bone, is only visible thanks to a ridge that marks its anterodorsal margin. The rest of the fossa is covered by matrix and by the dermal skull bone, which also prevents seeing whether a sutural surface for the prearticular is present or not.
- An isolated bone, squarish to ovoid in shape, is attached by matrix to the posterior part of the angular. The natural margins of the bone are not well preserved, preventing

its precise identification. Its surface is ornamented with a reticulation of bulbous ridges, pretty different from the diverging ridges of the angular. This difference in ornamentation between the skull roof and cheek bones, including the angular bone, however, is typical of what is observed in mawsoniid coelacanths. Consequently, we consider that this ossification likely belongs to the same individual as the angular, and it was tentatively identified as a supraorbital because of its general rectangular shape and its ornamentation being more pronounced than the ornamentation on the bones of the median series (parietals, nasals).

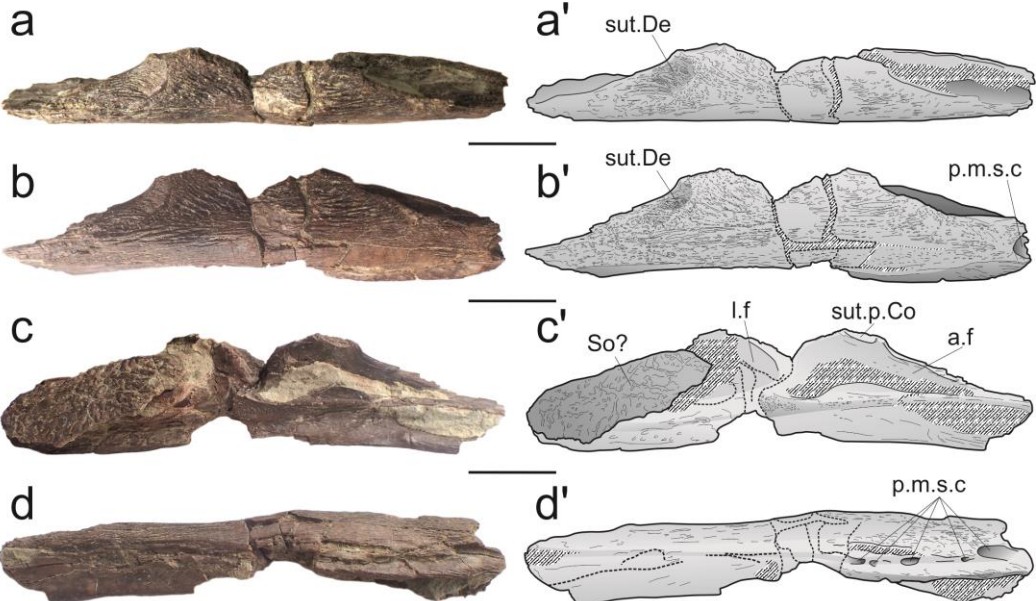

**Figure 2.** Photographs (**a**–**d**) and interpretative drawing (**a′**–**d′**) of the left angular of an indeterminate mawsoniid, Kham Phok locality, basal Cretaceous. Dorsal (**a**,**a′**), lateral (**b**,**b′**), internal (**c**,**c′**), ventral (**d**,**d′**) views.

### 4.1.2. Comparisons and Identification

The general shape and organization of the bone—localization of the fossa, sutural surfaces with surrounding bones, ornamentation, sensory canal—correspond without doubt to an angular of a coelacanth fish. Specifically, the bone can be referred to the *Mawsonia*/*Axelrodichthys* complex due to an ornamentation consisting of radiating ridges on the labial surface, a slightly inflated lateral surface and the presence of a sutural contact with the main coronoid. In both genera, the sensory canal openings are oval and few in number, but the situation is unclear in PRC 160. However, the preserved openings are relatively large and located ventrally, which is reminiscent of these genera.

As previously reported by Cavin et al. [22], the angular bone is a commonly preserved ossification for the *Mawsonia*/*Axelrodichthys* complex due to its robustness (Figure 3). Consequently, diagnostic characters have been commonly defined there, although it has always been difficult to separate diagnostic characters from intraspecific variations [22,32]. Based on a discussion of diagnostic characters by several authors [2,4,16,22,33], PRC 160 exhibits a mixture of characters, most of which are shared with *Mawsonia* (long overlapping surface with dentary, coronoid eminence only very slightly inclined forward, small sutural contact surface with principal coronoid), but there is also an important character shared with *Axelrodichthys*, the deepest point of the angular positioned at the anterior third of the bone. Several specific characters have been defined on angular ossification but, as summarized in Cavin et al. [22], most are debatable and possibly represent individual variation. In the current state, we prefer to relate these bones (angular and suborbital?) to an indeterminate mawsoniid belonging to the *Mawsonia*/*Axelrodichthys* complex.

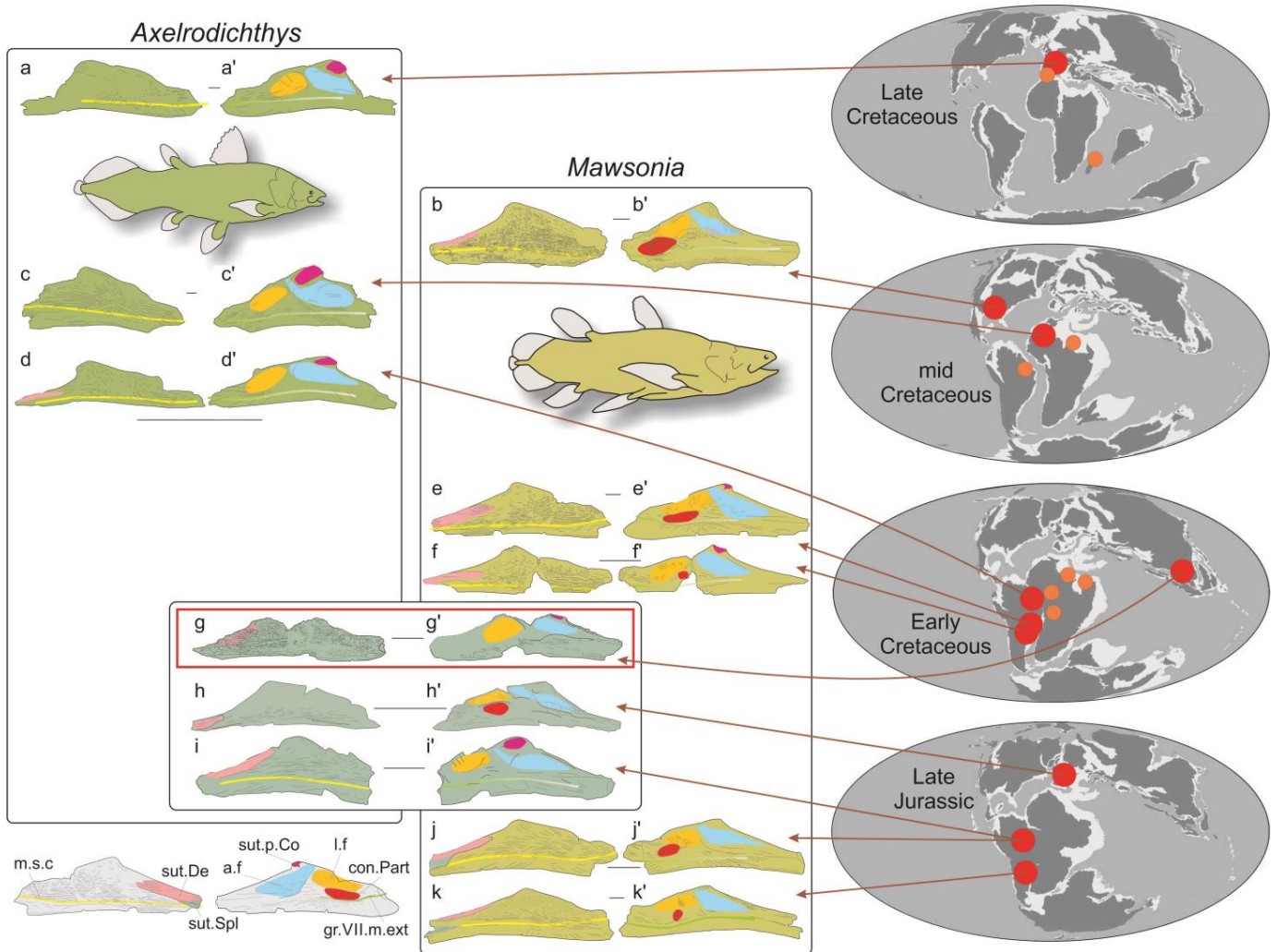

**Figure 3.** Comparison in labial (**a**–**k**) and lingual (**a′**–**k′**) views between left angular PRC 160, Kham Phok locality ((**g**), framed in red), and other angulars referred to the *Mawsonia/Axelrodichthys* complex, discovered in various parts of the world dated between the Late Jurassic and the Late Cretaceous (red spots). Remark: Specimens i and j, figured by Cupello et al. [4] and Batista et al. [5] are both from a formation called Missão Velha and Brejo Santo, respectively, with an uncertain age of the Late Jurassic or Late Cretaceous. (**a**) *A. megadromos*, Southern France, terminal Cretaceous (inverted); (**b**) *M.* sp., USA, Woodbine Fm. (inverted); (**c**), *A. lavocati*, Northern Africa, 'Continental Intercalaire'; (**d**) *A. araripensis*, Brazil, Santana Fm. (inverted); (**e**) *M. gigas*, Brazil, Marfim Fm.; (**f**) *M. gigas*, Brazil, Sanfranciscana Fm. (inverted); (**h**) mawsoniid indet., UK, Kimmeridge Clay; (**i**) mawsoniid indet., Brazil, MissãoVelha/Brejo Santo Fm.; (**j**) *M. gigas*, Brazil, MissãoVelha/Brejo Santo Fm. (inverted); (**k**) *M. gigas*, Brazil, Taruarembó Fm. Data from Cavin et al. (2021) [22]. Orange spots indicate other mawsoniid remains, not detailed here. The main anatomical structures are figured with colored areas. Abbreviations: a.f, adductor fossa (blue); con.Part, contact surface with prearticular (red); gr.VII.m.ext groove for external mandibular ramus of VII (green); l.f, longitudinal fossa (orange); m.s.c, mandibular sensory canal (yellow); sut.p.Co, sutural contact surface with principal coronoid (purple); sut.De: sutural surface for dentary (pink); sut.Spl: sutural surface for splenial (grey).

## 5. Discussion

The Mesozoic Asian (non-Middle Eastern) fossil record of coelacanths is poor. Three genera have been described from the Triassic of China [34,35], and two of them, *Luopingcoelacanthus* and *Yunnancoelacanthus*, have been resolved as basal mawsoniids in a recent phylogenetic analysis [36]. However, this phylogenetic position is questioned in a new cladistic

analysis in progress (C. Ferrante, personal communication 2022). *Indocoelacanthus*, from the Lower Jurassic Kota Formation in India, a landmass that was unconnected to mainland Asia at that time, is possibly a mawsoniid [16]. The only other Mesozoic Asian occurrence, to our knowledge, is *Whiteia oishii* from the Upper Triassic of West Timor, Indonesia [37]. The new occurrence described here therefore corresponds to the first coelacanth from the Cretaceous of Asia and the first coelacanth from Thailand.

Species of the *Mawsonia*/*Axelrodichthys* complex inhabited fresh or brackish water environments, with rare marine occurrence, discussed in Cavin et al. [38]. They were unlikely to cross large expanses of sea, and their distribution should largely reflect continental connections. First seen as a typical example of a West Gondwanan vicariant event associated with the opening of the South Atlantic Ocean in the Early Cretaceous [39], the biogeographical pattern of the *Mawsonia*/*Axelrodichthys* clade has become more complex with the discovery of European occurrence in the terminal Cretaceous, which involved dispersals during the Late Cretaceous [21,33]. Very recent discoveries in the 'mid' Cretaceous (Cenomanian) of North America [22] and Europe [23] further blur the signal and imply either earlier or multiple dispersal events or imply rethinking the palaeobiogeographical scenario of the entire clade. The discovery in the Lower Cretaceous of a member of this clade in Southeast Asia supports this last option.

In the most recent mawsoniid phylogenies [33,36], the sister genus of the *Mawsonia*/*Axelrodichthys* complex is the European Jurassic genus *Trachymetopon* (considering *Lualabea* as a member of the *Mawsonia*/*Axelrodichthys* complex, possibly a synonym of one of these genera [33]). *Trachymetopon* is one of the rare marine genera of which the stratigraphic range extends over the Jurassic, from the Toarcian to possibly the Tithonian [38,40,41] (but see below for possible reinterpretation of the Late Jurassic occurrences).

According to Dutel et al. [41], the angular of *Trachymetopon* is long and low, with a straight outline and coarse ornamentation formed by radiating ridges, a long overlapping surface with the dentary and the deepest point of the angular located approximately at the level of its mid-length, i.e., characters also present in *Mawsonia*. The situation becomes more complex with the very recent study of mawsoniid material discovered in the Upper Jurassic (Kimmeridgian) Kimmeridge Clay Formation of southern England by Toriño et al. [42] (Figure 3h). Various cranial elements belonging to the same individual have been referred to an indeterminate mawsoniid coelacanth related to the *Trachymetopon*/*Mawsonia*/*Axelrodichthys* group, but with more affinities with *Mawsonia*, in particular on the basis of characters present on the angular. This study also challenges the generic attribution of the Middle Jurassic occurrence of *Trachymetopon* from Normandy, France, which was previously referred to this genus partly for stratigraphic and geographical reasons [38,40,41].

The stratigraphic ranges of *Axelrodichthys* and *Mawsonia* were found to be very long, i.e., several tens of millions of years each [22], a situation that reflects a very slow rate of evolution. An interesting line of research suggested by Toriño et al. [42] is that *Trachymetopon* and *Mawsonia*, to which *Axelrodichthys* can be added here, may represent chronotaxa within one general lineage for over 80 million years. If true, this may explain the difficulty of drawing boundaries between genera and between their constituent species. In addition, species distinction is made more difficult by the presence of strong morphological variations within a single population of *Mawsonia gigas* from the Lower Cretaceous of the Sanfranciscana Basin, Brazil [32], a situation also observed in other populations of mawsoniids (LC, personal observation).

The comparison with the extant *Latimeria* is instructive, although no fossils referred to this genus have ever been found. Like the Jurassic–Cretaceous mawsoniids, *L. chalumnae* presents intraspecific polymorphism, notably in the pattern of the cheek bones [16], which is incidentally a skeletal module bearing characters considered diagnostic in fossil coelacanths. Although the osteology of the Indonesian coelacanth, *L. menadoensis*, is still incompletely known, the differences recorded between the two species are small and concern mostly morphometric and meristic values [43], perhaps not different enough to distinguish

separate species without the use of genetic tools. Finally, genetic data indicate a divergent time of the two species of *Latimeria* between 30 and 40 million years ago [44], indicating a very slow rate of morphological evolution over tens of millions of years, a situation quite similar to the situation observed for the Jurassic–Cretaceous mawsoniids.

Due to the slow morphological evolution within the mawsoniid coelacanth lineage, for potential biological reasons discussed by Cavin & Alvarez [45], we hypothesize that their evolutionary history might be disconnected, at least in part, from the palaeogeographical framework in which they lived. We suggest that different lineages within the *Mawsonia/Axelrodichthys* clade evolved independently in different parts of the fragmenting Pangea, but we are simply unable to recognize them as they are almost indistinguishable based on osteological characters due to their slow evolution. In short, because the paleogeographic evolution would be faster than morphological evolution of the mawsoniids, the evolutionary (phylogenetic) pattern would seem to be unrelated to their geographical distribution over time.

## 6. Conclusions

The discovery of a mawsoniid coelacanth in the Lower Cretaceous of Thailand is an important new addition to the already rich vertebrate assemblages of the Phu Kradung Formation and, more generally, the Jurassic–Early Cretaceous assemblages of the Khorat Group. On a global scale and on the scale of tens of millions of years, this new occurrence blurs the palaeobiogeographical model previously proposed for this clade. It questions both the vicariance and dispersal events previously proposed to explain the observed distribution. The recognition of the coelacanths as forming a slowly evolving clade [45], in particular the mawsoniid clade, can explain the disconnection between the phylogenetic pattern and the paleobiogeographical framework, whereas such a connection is normally expected in biogeographical studies. One way to test this scenario is (1) to attempt to build a stronger phylogeny based on a re-study of known and hopefully new material and (2) to better decipher the Cenozoic evolutionary history of the extant coelacanth, *Latimeria*, which shows a split of species between 30 and 40 million years ago associated with almost no morphological differentiation and therefore represents a good model to better understand the Cretaceous mawsoniid case.

**Author Contributions:** Conceptualization, L.C.; methodology, L.C.; software, L.C.; validation, L.C., H.T., E.B., K.W., V.S. and U.D.; formal analysis, L.C.; investigation, L.C., H.T., E.B., K.W., V.S. and U.D.; resources, L.C., H.T., E.B., K.W., V.S. and U.D.; data curation, U.D.; writing—original draft preparation, L.C.; writing—review and editing, L.C., H.T., E.B., K.W., V.S. and U.D.; visualization, L.C.; supervision, L.C.; project administration, H.T., E.B., K.W., V.S. and U.D.; funding acquisition, L.C., H.T., E.B., K.W., V.S. and U.D. All authors have read and agreed to the published version of the manuscript.

**Funding:** This paper is a contribution to the project 'Burst and Stasis in morphological evolution of Mesozoic coelacanths' supported by the Swiss National Science Foundation (200021_207903).

**Institutional Review Board Statement:** Not applicable.

**Data Availability Statement:** Not applicable.

**Acknowledgments:** This work was supported by a grant from the Swiss National Science Foundation (SNSF) and the International Research Group PalBioDivASE (IRN) grant of CNRS. We thank Christophe Ferrante for information about a new phylogeny of coelacanths, as well as the three anonymous reviewers for their constructive comments.

**Conflicts of Interest:** The authors declare no conflict of interest.

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
