# Peer review of "The First Fossil Coelacanth from Thailand"

_diversity, doi:10.3390/d15020286_

Round 1
Reviewer 1 Report
Strong ornamentation and wider sensory pores in the angular are also characteristics of the genus Mawsonia, if you see these characters, we suggest that they be mentioned in the text.
Author Response
- Comment: “Strong ornamentation and wider sensory pores in the angular are also characteristics of the genus Mawsonia, if you see these characters, we suggest that they be mentioned in the text.”
- Reply: we agree with this comment, but we consider that we already recognize this point in our original submission: “Specifically, the bone can be referred to the Mawsonia/Axelrodichthys complex due to an ornamentation consisting of radiating ridges on the labial surface, a slightly inflated lat-eral surface, the presence of a sutural contact with the main coronoid. In both genera the sensory canal openings are oval and few in number,…”
Reviewer 2 Report
This article describes the discovery of an angular and a possible supraorbital of a Mawsoniidae coelacanth from the Lower Cretaceous of northeastern Thailand, and discusses, succinctly, the paleogeographic distribution of the clade, the idea of this clade having evolved independently in different parts of Pangea, as well as the idea of a slow morphological evolution within this group.
The description is clear and well written and, by itself the discovery of a mawsoniid in the Lower Cretaceous of Thailand deserves to be published.
I have just some very few comments below.
- In the last paragraph of the introduction you use the term basal Cretaceous. Wouldn't it be better to replace by Lower Cretaceous?
- Regarding the M. gigas and the mawsoniid indet. from the Brejo Santo Formation (Araripe Basin). This geological Formation (Brejo Santo) is the same as the Missão Velha Formation sensu Cupello et al. (2016) and many other authors (Martill, Brito, etc). Therefore, the material described for exemple by Batista et al. (2019) is exactly the same (even some same specimens), collected in this unique locality, and its Jurassic age deserves more caution. I would suggest a ?Jurassic age or just a "Wealdian"age.
In summ, I consider this article should be published and I hope these minor comments would help.
Sincerely.
Author Response
- Comment: In the last paragraph of the introduction you use the term basal Cretaceous. Wouldn't it be better to replace by Lower Cretaceous?
- Reply: Done
- Comment: Regarding the M. gigas and the mawsoniid indet. from the Brejo Santo Formation (Araripe Basin). This geological Formation (Brejo Santo) is the same as the Missão Velha Formation sensu Cupello et al. (2016) and many other authors (Martill, Brito, etc). Therefore, the material described for exemple by Batista et al. (2019) is exactly the same (even some same specimens), collected in this unique locality, and its Jurassic age deserves more caution. I would suggest a ?Jurassic age or just a "Wealdian"age.
- Reply: We added in the caption of figure 3 the following sentence: Remark: Specimens i and j, figured by Cupello et al. [4] and Batista et al. [5] are both from a formation called Missão Velha and Brejo Santo, respectively, with an uncertain age of Late Jurassic or Late Cretaceous.
Reviewer 3 Report
The paper is a simple one in that it reports a new fossil of a coelacanth fish from the Cretaceous of Thailand, the first record of the group from that nation and region. The introduction and description of the fossil is excellent and professional, and the discussion sets the diascovery in context in a very useful and comprehensive manner with a thorough overview of the whole issue of the genera Mawsonia and Axelrodicthys and difficulties of discriminating the two genera, as well as paleobiogeographic implications.
My only revision suggestions concern language – I list a few, but ask the authors please to set up Word with ‘Tools/Language = English (US)’ and untick the ‘do not check’ box, so you get complete spell and grammar check. This would have picked up some of the points I make, and might find a few other awkward phrases and mis-spellings you can correct.
Figure 2: the reproduction of the bone photographs nis quite dark – can the color be lightened a little?
Figure 3: typesetter – please organise the placement of the image so the whole caption fits on the same page please.
Language
Page 2: leading to hypothesize = leading to an hypothesis of
Distribution biogeography = biogeographical distribution
Page 3: except some breaks = except for some breaks
Page 4: wether = whether
Page 8, ref. 14: remove un-needed capital letters in paper title
ref. 21: remove un-needed capital letters in paper title
Page 9, ref 22: cretaceous = Cretaceous
ref. 28: T hailand = Thailand
ref. 45: remove un-needed capital letters in paper title
Author Response
- Comment: My only revision suggestions concern language – I list a few, but ask the authors please to set up Word with ‘Tools/Language = English (US)’ and untick the ‘do not check’ box, so you get complete spell and grammar check. This would have picked up some of the points I make, and might find a few other awkward phrases and mis-spellings you can correct.
- Reply: Done
- Comment: Figure: Figure 2: the reproduction of the bone photographs nis quite dark – can the color be lightened a little?
- Reply: Done
- Figure 3: typesetter – please organise the placement of the image so the whole caption fits on the same page please.
- Reply: Done
Language Reply: Done